# Oral and Parenteral Vaccination against *Escherichia coli* in Piglets Results in Different Responses

**DOI:** 10.3390/ani12202758

**Published:** 2022-10-14

**Authors:** Guillermo Ramis, Lorena Pérez-Esteruelas, Carolina G. Gómez-Cabrera, Clara de Pascual-Monreal, Belén Gonzalez-Guijarro, Ester Párraga-Ros, Pedro Sánchez-Uribe, Miguel Claver-Mateos, Livia Mendonça-Pascoal, Laura Martínez-Alarcón

**Affiliations:** 1Departamento de Producción Animal, Facultad de Veterinaria, Universidad de Murcia, 30100 Murcia, Spain; 2Instituto Murciano de Investigación en Biomedicina (IMIB), 30120 Murcia, Spain; 3ELANCO Animal Health, 28108 Alcobendas, Spain; 4Departamento de Anatomía y Anatomía Patológica Comparadas, Universidad de Murcia, 30100 Murcia, Spain; 5Escola de Veterinária e Zootecnia, Universidade Federal de Goiás, Goiánia 74690-900, Brazil; 6UDICA, Hospital Clínico Universitario Virgen de la Arrixaca, 30120 Murcia, Spain

**Keywords:** *Escherichia coli*, vaccine, oral pathway, parenteral pathway, biomarkers

## Abstract

**Simple Summary:**

One of the strategies for the prevention of *E. coli* related problems is the vaccination of piglets. Vaccines with different routes of administration are available: oral and parenteral. The former mimics the natural route of infection. Two different responses have been defined depending on the route of administration, with differences being observed in the number of IgA-producing cells, cytokine activation and intestinal integrity, depending on the route used. In general, there is evidence of greater immune system activation in the orally vaccinated group, which may indicate that the parenterally vaccinated group needs a second *E. coli* stimulus to fully develop the immune response. It should be noted that this is not an efficacy study as the animals were not inoculated and did not suffer from clinical problems related to *E. coli*.

**Abstract:**

The available *E. coli* vaccines involve two main types (inactivated and live non-pathogenic) and two routes of administration (oral and parenteral) but the mechanism by which both vaccines and routes of administration work is not yet fully elucidated. The influence of a parenteral vaccine (PV) and an oral one (OV) was studied by analyzing the gene expression of biomarkers indicating cellular infiltration (calprotectin, CAL), tight junction proteins (occludin OCL, and zonulin ZON) that maintain intestinal paracellular integration and two proinflammatory (IFN-γ) and anti-inflammatory (TGF-β) mediator cytokines, as well as histomorphology and IgA production cell density. Differences were observed in CAL, more infiltrated in orally vaccinated animals; OCL also increased in orally vaccinated animals, and higher density of IgA-producing cells in ileum for orally vaccinated groups. Cytokine expression is also different; and there is a lower mRNA for IFN-γ in the parenteral than in the oral vaccinated animals. Finally, the villus height-to-crypt depth ratio was higher in the orally vaccinated groups. The data collectively show clear and different effects derived from the use of each type of vaccine, route of administration and regimen. The results suggest a more rapid and direct effect of oral vaccination and a state of suppression in the absence of a second oral stimulus by the pathogen.

## 1. Introduction

*Escherichia coli* is a ubiquitous, universally distributed pathogen responsible for a large proportion of diarrhea in piglets, both neonatal and post-weaning diarrhea (PWD) [1]. The pathogenic action of *E. coli* depends on the adhesion elements present, as well as the ability to produce thermolabile toxins (LT), thermostable toxins (Sta and Stb) and verotoxins (VT1 and VT2) [1,2]. In fact, *E. coli*-related diseases may be considered one of the most economically costly diseases in the pig industry and, from a public health point of view, one of the most antibiotic-intensive in recent years.

For decades, prevention against this pathogen has been carried out by vaccinating gilts and sows to boost colostral immunity [3], using quantities of antibiotics such as colistin and adding zinc oxide at therapeutic levels in the feed [4], which means the use of 3000 ppm of this element through the feed. However, the latter two strategies have been abandoned in the European Union [5,6], the first, because colistin has been declared a last resort in human medicine given the increased resistance to the most commonly used antibiotics and because colistin has classically been under-prescribed in human medicine. Additionally, the use of zinc oxide has been banned without the possibility of a moratorium in June 2022 due to environmental problems resulting from the accumulation of this metal in areas where pig manure is used as an organic amendment in agriculture or where there is accidental or intentional dumping of manure. The enteric health situation in piglets has worsened markedly in the last 5 years, with increasing outbreaks of diarrhea and mortality in nurseries [7]. Not only has there been a worsening of piglet intestinal health, but there has been a re-emergence of diseases such as porcine streptococcal disease caused by *Streptococcus suis*, which could be due to a loss of intestinal integrity as proposed in a model of intestinal translocation of Streptococcus suis bacteria [8].

Another prevention alternative is the direct vaccination of piglets, which will only be useful in post-weaning diarrhea due to the time needed for the vaccines to produce an adaptive response. Most strains that produce PWD in piglets present F4 and/or F18 fimbriae, so vaccines usually contain at least these two adhesion elements. However, there is a great heterogeneity of adhesion or virulence factors among the *E. coli* strains present in the populations. This heterogeneity is what makes it difficult to prevent PWD, since if the pathotypes present in a herd do not express the adhesion factors contained in the vaccine administered, there will be no blockage of pathogen adhesion, which is essential to produce effective immunity [9].

Currently, both parenteral and oral vaccines are available; the former are usually based on inactivated bacteria and the latter on live apathogenic strains of the bacterium, but not many specific brands are commercially available, at least in Europe. This does not mean that there are a significant number of publications on testing oral, parenteral and intranasal delivered candidate vaccines, based on live attenuated strains chimeras, purified or synthetic subunits (fimbriae and toxins) or RNA [2,9,10,11,12,13]. Among the available oral vaccines are monovalent or bivalent vaccines including strains containing F4 and F18 adhesion factors. Inactivated vaccines usually include several strains that ensure the presence of different adhesion elements, as well as thermostable and thermolabile toxins and shiga-like toxins. Even *Clostridium perfringens* toxoids are often included to provide simultaneous mixed protection against both pathogens. From the point of view of management and use of the vaccine, each type has advantages and disadvantages. Oral vaccinations are less invasive than parenteral vaccinations as they do not involve the use of needles, whether piglets are vaccinated by drinking water or by direct drenching. Parenteral vaccinations have the disadvantage of the management required, the possibility of iatrogenic transmission or the metal waste generated by discarded needles. Nevertheless, they have the great advantage of ensuring that all animals receive a full dose, which is not assured when oral vaccines are administered by drinking water.

However, the ability to produce local secretory IgA-based immunity in the gut by administering vaccines parenterally, subcutaneously, intradermally, intradermally, intraperitoneally or transdermally, not only in pigs but also in humans, is still under discussion [14]. Processing of orally administered antigens via the gut associated lymphoid tissue (GALT) produces a local response based on these immunoglobulins which, in the case of *E. coli*, should provide protection against post-weaning diarrhea if antibodies are produced against the adhesion elements or exotoxins produced by the bacteria. However, systemic processing of parenterally administered antigens could produce a systemic serological IgA response that does not correspond to the local response necessary for gut protection, since memory B cells stimulated against *E. coli* antigens would have to be present in the GALT at the time of infection [9].

Despite the availability of these vaccines on the market for years, the immune mechanisms that trigger both types of vaccine and both routes of administration have not yet been fully elucidated. Similarly, there is little information available on the effect of short-term vaccination on intestinal integrity, understood as the integrity of tight junctions and other paracellular junctional elements of the intestinal epithelium, a key element of defense against enteric diseases.

In this study, the effect of the use of an oral and a parenteral vaccine on gene expression of tight junction proteins as intestinal integrity biomarkers, gene expression of two of the key cytokines involved in the vast majority of vaccine responses, IgA production cell density and intestinal morphology in post-weaned piglets was compared.

## 2. Materials and Methods

### 2.1. Animals

Seventy large white × Landrace piglets from a farm without clinical problems related to *E. coli* were used to avoid interference with the effect of the vaccine. The previous diagnosis showed the presence of F4-*E. coli* in rectal swab samples obtained from a total of 30 animals. Swabs with fecal matter were inserted into Amies transport medium in airtight screw cap plastic vials and were submitted to the laboratory (Exopol; Zaragoza) for *E. coli* diagnosis. A multiplex PCR was carried out individually to detect adhesion factor genes, including F4 (K88) and F18 fimbriae. The animals were slaughtered 21 days after the oral vaccination. All the experimental procedures described in this research were carried out under the welfare rules stated in R.D. 1135/2002 for the protection of pigs and the trial was approved by the Bioethical Committee of the University of Murcia (CEEA-OH 465/2018).

The trial was developed in the Veterinaria Teaching Farm of the University of Murcia. The piglets were weaned at 22 ± 1 days of life and placed in flat deck pens containing 9 piglets each. The experimental groups were separated by room to avoid transferring alive vaccine strain from one to the other in case of excretion. Personnel flow was CON-PV-OV and never in the opposite direction.

### 2.2. Vaccination

The piglets were vaccinated with two different vaccines: on one hand, a parenteral vaccine (Colidex-C, Vetia, Spain) containing inactivated *E. coli* strains P6, P1, P2, P4, P5, P9 and P10 which ensures the presence of adhesins F4ac, F5, F6, F41, F18ab, F18ac and toxoid β from *Clostridium perfringens* type C. On the other hand, it was used a vaccine via oral, Coliprotec F4/F18 (Elanco GmbH, Monheim am Rhein, Germany), containing *E. coli*, alive non-pathogenic strains O8:K87 (1.3 × 10^8^ to 9 × 10^8^ UFC) and O141:K94 (2.8 × 10^8^ to 3 × 10^9^ UFC) assuring the presence of adhesins F4ac and F18ac. The piglets were vaccinated at 10th (0.5 mL) and 20th (1 mL) days of life for PV group (n = 20), and at weaning (OV; n = 20) for oral vaccine. After being rehydrated in mineral water, the vaccine was delivered by drenching to the piglets; 2 mL.

Thirty piglets were left as controls (CON); ten piglets received two saline injections in the cervical area at the same time of vaccination with PV, and ten piglets received one oral saline pulse at weaning, and ten piglets received nothing. None of the piglets were treated with antimicrobials into the 21 days after vaccination.

Five animals were housed in each pen, with six replicates of control animals, four replicates of OV animals and four replicates of PV animals.

### 2.3. Growth Performance, Clinical and Pathological Examination

The animals were individually weighted at weaning and by the time of necropsy.

Piglets were monitored daily for general health and fecal observation during lactation; before vaccination, and for 3 weeks after the oral vaccination. The fecal score in a pen basis was observed every two days during finishing, according to a 1 to 3 score: 0, normal feces, 1 pasty, 2 creamy and 3 liquids. The score was assigned to the highest score observed regardless of the number of animals showing this score.

### 2.4. Sampling

Animals were humanely killed by intravenous thiobarbital overdose, and immediately followed by necropsy, with assessment of the presence of lesions in the organs with special attention to the digestive tract. Finally, a sample of jejunum, ileum and colon were fixed in 4% formalin. An adjacent 100 mg portion was preserved in RNAlater^®^ (Invitrogen, Waltham, MA, USA), for subsequent RNA isolation.

### 2.5. RNA Isolation and cDNA Synthesis

Briefly, total RNA was isolated from 20 mg of tissue samples by using the Thermo Scientific Gene JET RNA Purification Kit (ThermoFisher, Waltham, MA, USA) and cDNA was synthetized using the Geneamp RNA PCR Core Kit (Life Technology) using oligo-dT as primers to get cDNA only from mRNA.

### 2.6. Gene Expression for TJ Protein and Cytokines

Gene expression was investigated for two tight junction (TJ) proteins: occluding (OCL) and zonulin1 (ZON), an indicator of inflammatory cells presence: calprotectin S100 (CAL); protein present in neutrophils and macrophages, and two cytokines as representative of pro-inflammatory activity (IFN-γ) and anti-inflammatory activity (TGF-β). The primers for each gene are shown in Table 1, previously published primers were used for IFN-γ, TGF-β and β-actin, and designed using PrimerBlast (https://www.ncbi.nlm.nih.gov/tools/primer-blast/ (accessed on 25 June 2021) those for CAL, ZON and OCL using the sequencies referred in the Table 1.

The PCRs were performed using a 7300 ABI thermocycler (Life Technologies, Carlsbad, CA, USA) and the GoTaq q-PCR Master Mix (Promega, Madison, WI, USA) with SYBR-Green chemistry. The specificity of the reaction was assessed by analyzing the melting curve. The samples were normalized using the Ct for β-actin. The expression for each sample was calculated, correcting to the PCRs efficiency, which was calculated by serial decimal dilutions and using the slope offered by the thermocycler software, and used as control group the CON animals, as previously described [15], and using as housekeeper β-actin. Data were expressed as fold change, normalized to the lowest value (which was assigned a value 1), and then as log2 to compare the different treatments.

### 2.7. Histomorphology

Formalin-fixed tissues were routinely processed for paraffin embedding; 5 μm thick sections were obtained, and serial sections were made, stained with HE for morphometric measurements and for immunocytochemistry.

After staining, the slides were scanned with a PANNORAMIC 1000 slide digital scanner (3DHISTECH Ltd., Budapest, Hungary). The digitized slides were entered into the Slideviewer software (3DHISTECH Ltd., Budapest, Hungary) and 10 non-consecutive well-oriented villi were measured in the ileum and jejunum, as well as the depth of 10 crypts adjacent to these villi. The villus height-to-crypt depth ratio was calculated by dividing villus height by crypt depth. All morphometric measurements were performed by the same researcher, who was blinded to the treatments.

### 2.8. Immunohistochemistry

Consecutive sections of the samples in paraffin were obtained for the IgA immuno-cytochemistry study in jejunum, ileum and colon tissue.

The samples were dewaxed and dehydrated with graded ethanol and the endogenous peroxidase activity was quenched in 3% H_2_O_2_ in methanol for 30 min. Samples were pretreated with 10% pronase in TBS (Sigma-Aldrich, St. Louis, MO, USA) for antigen retrieval (12 min). Afterwards, the samples were rinsed in TBS for (3 × 5 min) and incubated for 30 min with 100 μL of blocking solution per slide at 20 °C in a humid chamber. Subsequently the samples were incubated for 1 h at 37 °C with the primary antibody (goat- anti-pig IgA, Bethyl, Montgomery, TX, USA) diluted 1:3000 in TBS. The secondary antibody (biotin conjugate rabbit anti-goat, Dako, Carpinteria, CA, USA), diluted 1:300 in TBS, was incubated for 30 min at 20 °C. The avidin–biotin-peroxidase complex technique was used with the Vectastain^®^ Elite ABC kit (Vector, Newark, CA, USA), which was applied for 1 h at 20 °C. Positive labeling was detected using 3,3′-diaminobenzidine tetrahydrochloride (Dako, Carpinteria, CA, USA). Sections were counterstained with Mayer’s haematoxylin, dehydrated and mounted. The number of IgA-producing cells in the intestinal lamina propria was counted using a Zeiss Axioskop 40 microscope (Carl Zeiss, Oberkochen, Germany) with a Spot Insight camera and the Spot Advanced software (Spot Imaging Solution, Michigan, USA). Immunolabeled cells were counted in 10 non-overlapping consecutive high magnification fields and expressed as amount of cells/25.000 μm^2^.

### 2.9. IgA Content

To study the IgA content in the tissues, the RNAlater was used in which the samples were preserved for RNA extraction, as previously described [19]. Prior to RNA extraction, the RNAlater was removed by centrifugation and preserved at −80 °C until processing. The commercial Pig IgA ELISA kit (Bethyl Laboratories, Montgomery, TX, USA) was used to determine the amount of IgA contained in the preservative. The RNAlater was centrifuged prior to use, 3900 *g* × 10 min, and the supernatant was diluted 1:2 with the kit’s sample diluent buffer. The ELISA was performed according to the manufacturer’s recommendations.

### 2.10. Statistical Analysis

All analyses were carried out using the statistical software package SPSS v. 23 (SPSS Inc., Chicago, IL, USA). The normality of the data was tested using a Kolmogorov–Smirnov test. Since the data were non-normal, a Kruskal–Wallis test was used for non-parametric comparison of multiple samples, with a Mann–Whitney U test for two-to-two comparisons of independent samples. A discriminant function analysis was used to perform a data reduction, considering only the functions for Wilks lambda *p* < 0.05. Two discriminant function analyses (DFA) were performed; the first one considered biomarker variables (CAL, OCL, ZON, TGF-β and IFN-γ) and quantification of IgA-producing cells but not histomorphometry as it was not performed in the colon. On the other hand, a second DFA was performed using the histomorphometry and IgA-producing cell data for jejunum and ileum. A group membership assignment analysis and a dot plot were performed to graphically represent the separation of groups when at least two canonical functions were obtained.

## 3. Results

### 3.1. Growth Performance and Clinical Examination

The weaning weight was 5.57 ± 0.42 Kg; 5.86 ± 0.43 and 5.25 ± 0.37 Kg for CON, PV and OV, respectively without significant difference among them (*p* = 0.585). At necropsy, the animals weighted 12.38 ± 0.82, 13.95 ± 1.12 and 13.02 ± 0.60, with an average daily gain of 0.27 ± 0.02; 0.33 ± 0.03 and 0.3 ± 0.03 for each treatment, again with no significant differences for weight (*p* = 0.386) or growth (*p* = 0.424). There were no differences among replicates for each treatment.

None of the piglets included in the study showed signs of diarrhea or general disease; and in fact, none of the piglets in any of the pens were observed to have feces with an assessment of 2 or 3 throughout the observation period.

### 3.2. Gene Expression for TJ Proteins and Cytokines

Relative gene expression results are shown in Figure 1, as log2 of the fold change, expressed as mean ± SEM.

Regarding CAL, the highest quantification was always in the OV group, while the lowest gene expression was always observed in the PV group. This could be explained by the type of vaccine (inactivated vs. live non-pathogenic) and the route of administration (parenteral vs. oral). Attending to the TJ proteins, it is interesting to note that while OCL has the highest quantification in the OV group and the lowest in the PV group, for ZON quantification, the exact opposite is true; the highest quantification occurs in the CON group and the lowest in the OV group. Cytokines showed an interesting quantitation; the PV group had significantly lower IFN-γ RNAm quantification than the CON and OV groups in jejunum and ileum, with no differences between groups in the colon. Regarding TGF-β, the OV group had significantly higher RNAm quantification than the CON and PV in the ileum and the PV group had significantly lower RNAm quantification than the CON and OV in the colon.

There were no differences among the CON animals in any of the measures.

### 3.3. Histomorphology

The results for villus height and crypt depth appear in Table 2. An example of images appears in Figure 2.

Morphometry results are consistent with other groups studied on the same farm [20].

The highest villus were observed in the PV group in the jejunum and for OV in the ileum. The highest depth for crypts was recorded for OV and PV groups in the jejunum and ileum, respectively.

Regarding the villus height-to-crypt depth ratio, the values appears in Figure 3. Significant differences were found between the control and PV groups compared to the OV group (*p* = 0.001 for CON vs. OV, *p* = 0.016 for PV vs. OV).

### 3.4. IgA-Producing Cells

The results for IgA-producing cells counting appears in Figure 4, and an example of high and low cell density in Figure 5. Significant differences were observed between the OV group with the CON and PV groups in ileum and colon (*p* < 0.0001 and *p* = 0.005, respectively). The increase in the number of IgA-producing cells conforms to an exponential trend line (r = 0.878, R = 0.956 and r = 0.987; *p* < 0.0001).

### 3.5. IgA Content

The results for IgA quantification in tissues are shown in Figure 6.

There were no significant differences in the quantification obtained in jejunum between the three treatments, while in the ileum and colon there were differences between the CON vs. OV group (*p* = 0.001 and *p* = 0.007 for the ileum and colon, respectively) and the PV group vs. OV group (*p* < 0.0001 and *p* = 0.001 for the ileum and colon, respectively).

When comparing quantification between tissues, the CON group showed differences between the jejunum and ileum (*p* = 0.007) and between the ileum and colon (*p* = 0.022); in the PV group, differences were observed between the jejunum and ileum (*p* = 0.046) and between the jejunum and colon (*p* = 0.022); and in the OV group, differences were observed between the jejunum and ileum (*p* < 0.0001) and between the ileum and colon (*p* = 0.001).

Correlations were obtained between IgA quantification, the number of IgA-producing cells (r = 0.155, *p* = 0.027), CAL (r = 0.257, *p* < 0.0001), OCL (r = 0.272, *p* < 0.0001), with TGF-β quantification (r = 0.203, *p* = 0.003), crypt length (r = −0.304, *p* < 0.0001) and the villus-to-crypt ratio (r = 0.289, *p* < 0.0001). When the correlation between IgA quantification and the number of IgA-producing cells is calculated without taking into account the jejunum samples, given that there are no differences between groups, a correlation of r = 0.317 (*p* = 0.006) is obtained.

### 3.6. Data Reduction

Finally, two functions were calculated, explaining 100% of the variance in the three tissues, with *p* < 0.0001 for Wilk’s lambda and the two functions, which means that the effect of vaccinations is clearly differentiable by taking mRNA biomarkers and quantification of IgA-producing cells. The concordance of sample assignment to groups depending on the calculated functions is shown in the Table 3.

For the jejunum, 68.8% of the original clustered cases were correctly classified; for the ileum, 72.0% were correctly classified; and for colon, 78.5% were correctly classified. The highest rates of correct asymmetry were always in the vaccinated groups.

The Figure 7 shows the canonical plots for discriminant functions.

When discriminant functions analysis was performed also considering histomorphology together with biomarkers and the number of IgA-producing cells, a Wilks’ lambda *p* < 0.001 was obtained with two functions explaining 96.8% and 98.3% of the variance in the jejunum and ileum, respectively. The concordance of sample assignment to groups depending on the calculated functions is shown in the table below (Table 4), and the DFA plot in the Figure 8.

For the jejunum, 79.5% of the original clustered cases were correctly classified and for the ileum, 73.6% were correctly classified. Again, the highest rates of correct asymmetry were always in the vaccinated groups.

When this analysis is performed using only the biomarker data, a Wilks’ lambda *p* < 0.0001 is obtained for the three tissues and a discrimination capacity of 70%, 64% and 75.8% of the cases, respectively. However, when discriminant function analysis is performed taking only histomorphology data and the number of IgA-producing cells, a Wilks’ lambda *p* = 0.017 for the jejunum and *p* < 0.0001 for the ileum is obtained, and a correct group assignment ability of 50.7% and 63.9% respectively for each tissue. This difference in discriminating ability for jejunum suggests that the effect of the vaccine is more marked on markers of integrity, immune stimulation and infiltration than on tissue morphology.

## 4. Discussion

The efficacy of piglet vaccination against *E. coli*, both orally and parenterally, has been demonstrated [21,22]. However, the different stimulations due to the two different delivery pathways has not been fully described. There is limited information on this in the scientific literature, and there is usually a short time period from the time of vaccination, including in other species such as mice. In this work, we assessed different statuses for biomarker gene expression, IgA production cell density and histological morphology 3 weeks after the last vaccination program compared to control unvaccinated animals reared in the same conditions.

Calprotectin is a heterodimer or homodimer of subunits S100A8 and S100A9, expressed by myeloid cells, especially neutrophils and monocytes [23]. The usefulness of this biomarker as an indicator of inflammation in various pathologies has long been recognized [24] and is well correlated with intestinal wall thickening [25]. It is usually investigated by fecal presence of the protein. However, the mRNA gene expression has also been investigated in experimental infections [26]. The authors researched the level of mRNA in Salmonella infected piglets, finding no important variations, which suggest a redistribution of the macrophages and/or neutrophils from the Peyer’s patches in close proximity to the lumen. However, in this work we found large variations for the mRNA of this marker, with a significant increase in the animals of the OV group compared to the other groups. It has been previously shown in mouse and pig models that parenteral immunization with inactivated F4ac strains via subcutaneous or intramuscular routes can induce a state of local specific suppression until secondary oral stimulation occurs [27]. This could explain the significantly lower CAL mRNA quantification in the PV group 21 days after the last vaccination.

Based on the quantification of IFN-γ and TGF-β, both of them increased in the OV group compared to the other groups. This result would indicate an active state of immune activation at the time of sampling. The difference with PV could be explained by the suppressed state mentioned above. An increase in secretion of Th1 cytokines such as IFN-γ and TNF-α has been shown previously following stimulation with F4 antigen alone or with adjuvants, which is consistent with our observations, as the OV group has a higher mRNA amount for this cytokine compared to the other groups [28].

An increase in ZON and OCL has been observed in post-weaned piglets by dietary interventions, which is interpreted as an increase in enterocyte differentiation and proliferation [29]. In our study, we observed that ZON is significantly decreased in the OV group compared to the others. In cell culture, it has been shown that the presence of IFN-γ and TNF-α can lead to the reorganization of TJ proteins and a loss of intestinal integrity [30]. In this study, groups with higher IFN-γ gene expression have lower ZON expression, although no significant correlation was found between the two markers in any tissue. However, it is plausible that there is a reduction in gene expression of the protein as a result of oral vaccination derived from the cytokines’ activation.

Increased villus height has been shown to be associated with improved intestinal health using various additives [31]. In terms of the villus height-to-crypt depth ratio, the orally vaccinated group showed a higher ratio in the ileum. Again, this may be due to the direct mucosal effect of the oral vaccine. It should be noted that the animals were slaughtered at least 21 days post-weaning, which means that there has already been a recovery of the weaning-related reduction in villi height, as demonstrated above [20,32]. Some preventive elements produce variations in the height of the villi but no variation in the depth of the crypts [31] or even no variations in measures compared to a control group [33]. We found variation in both parameters, as well as in the V/C ratio, in agreement with others using other strategies such us microbiota modifications [34]. It should be borne in mind that in this work we have used the classical 2D histomorphological analysis approach, which is the one used in most of the scientific works consulted. However, this approach has the disadvantage of requiring the sample to be sliced, which reduces a 3D reality to a plannar representation and therefore, for instance, surfaces become lines, and lines becomes points. In general, plannar representations of 3D objects do not have the sizes, shapes and numbers of the objects themselves [35,36]. In addition, one has to be sure that the chosen sample is representative of the total tissue. All this could be a drawback when making measurements of villi and crypts in 3D tissue using quasi 2D slices. Due to awareness of this limitation in the results obtained, we will approach this type of analysis in future research using uniform random sampling to choose the fields to be measured, as well as complementary stereological techniques to ensure a good estimation of the sizes of villi length and crypt depth, as currently used for studies of the digestive tract in various species [35,37,38]. The analysis of discriminant functions indicated a clear separation of treatments, both when carried out using biomarker gene expression results and when done using gut morphometry data. In the analysis carried out with the mRNA biomarkers, the allocation capacity in the vaccinated groups was high in the jejunum and ileum. The results obtained including histomorphology are comparable to those obtained with biomarkers and IgA-producing cell numbers alone. This would indicate that the effect of vaccination produces a similar effect on CAL, OCL, ZON, IFN-γ and TGF-β than on the histological structure. Probably, the mode of action of the orally administered vaccine, with direct stimulation of the intestinal mucosa, could result in a more evident effect on histomorphology, as suggested by our results.

Regarding quantification of IgA-producing cells, evidence that a mucosal IgA secretory response is required to confer protection against *E. coli* is well known. *E. coli* needs adhesion elements to colonize the small intestine. Most strains that cause post-weaning diarrhea have F4 and F18 fimbriae. These fimbriae allow the bacteria to proliferate and secrete exotoxins responsible for the pathogenesis of the disease. Therefore, the presence of secretory IgA against fimbriae on the surface of the gut at the time of infection is necessary to prevent E. coli adhesion [9]. Systemic vaccination, whether subcutaneous, intramuscular, intradermal, intraperitoneal or transcutaneous, can induce mucosal immunity depending on antigen/adjuvant combinations, but the mechanism that induces this protection is still poorly understood, even in humans [14]. Although a systemic IgA response has been demonstrated following parenteral vaccination in pigs, as well as an intestinal IgA response, effective vaccination against *E. coli* requires the presence of IgA-producing cells in the GALT at the time of infection. In both published trials with parenteral and oral vaccines, quantification of circulating IgA or IgA present in intestinal contents or tissue was carried out, but quantification of the density of IgA-producing cells in the different intestinal tracts has yet to be carried out.

An increase in the amount of circulating IgA against F4 and F18 has been demonstrated in the case of the oral vaccine [21], In piglets, an increase in IgA in the jejunum content has been observed compared to controls 10 and 24 days after vaccination [39]. In the case of parenteral vaccines, it has been shown that administration of F4 antigen together with vitamin D3 (as immunonodulator) at 7 and 23 days of age, with a subsequent challenge at 33 days of age producing a secondary IgA response after the challenge, suggesting that there were F4-stimulated memory cells in the GALT at the time of the challenge.

The fact that the oral vaccine produces significantly more IgA-producing cells in the mucosa of the ileum and colon compared to the CON and PV groups agrees with previous research which observed that oral vaccination, being the natural route of infection, induces a more evident local response than parenteral vaccination by any route. This is logical given that the oral route is the natural route of stimulation of IgA production as a local protective element against gastroenteric pathogens, since 90% of all pig pathogens invade through mucosa and therefore having these immunoglobulins is critical for the maintenance of health [28]. The fact that there is a difference in the ileum and colon but not in the jejunum could be due to the difference in GALT density between both tracts. Processing of *E. coli* antigens following oral vaccination with fimbrial subunits through M cells present in Peyer’s patches has been described [9]. In the case of the colon, we should not rule out that the vaccine strain proliferates with more ability in the colon, given that the posterior intestinal tract is the area of development of *E. coli* mainly.

When determining the amount of tissue-related IgA, it was observed that there was a significant difference in the amount of antibodies recovered from the tissue preservative in the ileum and colon, comparing the OV group with the CON and PV groups. In fact, in the ileum, which is the tissue where the highest IgA quantification is observed, there is a 35- to 350-fold increase when comparing the CON and PV groups with the OV group. The correlation between the amount of IgA recovered and IgA-producing cells is positive and significant but not very high (r = 0.155), suggesting that many of the cells stained in the tissues are not secretory. However, the differences in tissue IgA quantification occurred in the ileum and colon, precisely the same tissues where differences in the number of IgA-producing cells were observed between experimental groups. The presence of IgA in the ileum was between 10 and 50 times higher than the amount of antibody in the colon in the CON and OV groups, although a higher concentration of IgA-producing cells than in the jejunum or colon was not observed. Of interest is the positive correlation with the villus-to-crypt ratio and negative correlation with crypt depth; two of the indicators of intestinal integrity.

The results of this work suggest that PV produces a state of specific suppression until a second stimulus via oral route that would occur naturally by contact with the pathogen. This would be consistent with previously published data, in which parenterally vaccinated animals have a higher density of IgA-producing cells in the later stages of life than unvaccinated animals, which is not observed 21 days after the second vaccination in this study. This would corroborate the finding that a second oral stimulation is necessary in parenterally vaccinated animals [22]. However, oral vaccination would produce a direct mucosal stimulation with a higher density of IgA-producing cells at 21 days post-vaccination.

This work, especially those results showing a different immune response depending on the type of vaccine and route of administration, provides new information that may help to understand the response to these and other enteric pathogen vaccines. However, further research is needed to understand the gene expression patterns of other cytokines that may be involved in the immune response derived from *E. coli* vaccination by any of the pathways studied. In addition, intestinal integrity, in terms of TJ protein abundance, will be confirmed in the future by direct quantification of TJ proteins.

## 5. Conclusions

Vaccination with different vaccines and different routes of administration (oral and parenteral) results in different immune responses and variations in intestinal morphometry and IgA-producing cell density at 21 days post-vaccination. As expected, oral administration seems to produce a direct activation of local intestinal immunity immediately after vaccination, whereas parenteral vaccination seems to produce a suppressed state until oral stimulation by the bacteria occurs, which will be with the pathogen present in the populations.

## Figures and Tables

**Figure 1 animals-12-02758-f001:**
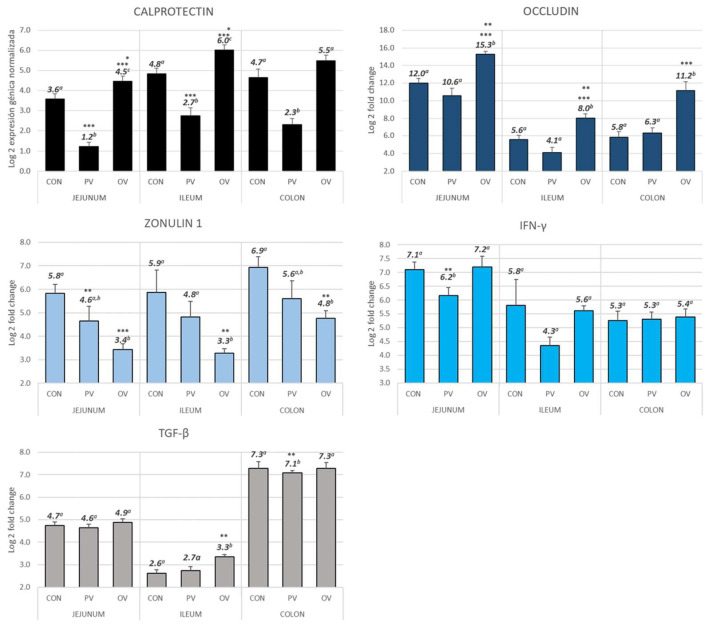
Gene expression for calprotectin, occludin, zonulin 1, IFN-γ and TGF-β, assayed in the jejunum, ileum and colon. The bars represent the mean ± standard error of the mean (SEM), and the data labels are the mean. Different superscript in each biomarker indicates significant differences. * indicates significance level *p* < 0.05, ** indicates *p* < 0.001 and *** *p* < 0.0001. When there are two indicators, the upper one marks the difference between CON and PV and the lower one between PV and OV. CON = control group, PV = parenteral vaccine group and OV = oral vaccine group.

**Figure 2 animals-12-02758-f002:**
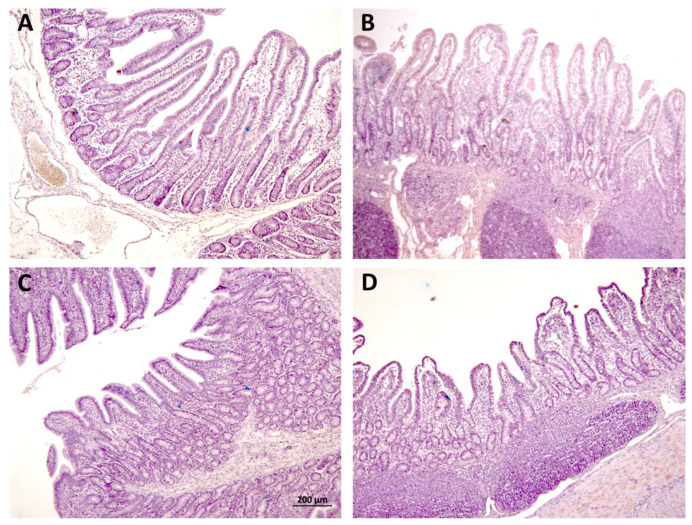
Example of high villus/short crypts in jejunum (**A**) and ileum (**B**), and short villi/long crypts in jejunum (**C**) and ileum (**D**). Hematoxylin–eosin stain. All the pictures have the same magnification; (×20). The bar size in the Figure (**C**) is 200 µm. a b c : the same shoulder letters mean no significant difference between groups; different shoulder letters mean significant difference between groups.

**Figure 3 animals-12-02758-f003:**
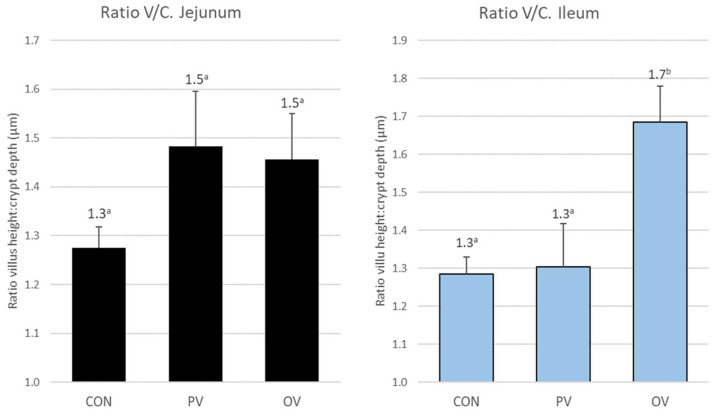
Villus height-to-crypt depth ratio in jejunum and ileum for each group. The bars represent the mean ± SEM. Different superscript in each biomarker indicates significant differences. The data labels are the mean. CON = control group, PV = parenteral vaccine group and OV = oral vaccine group.

**Figure 4 animals-12-02758-f004:**
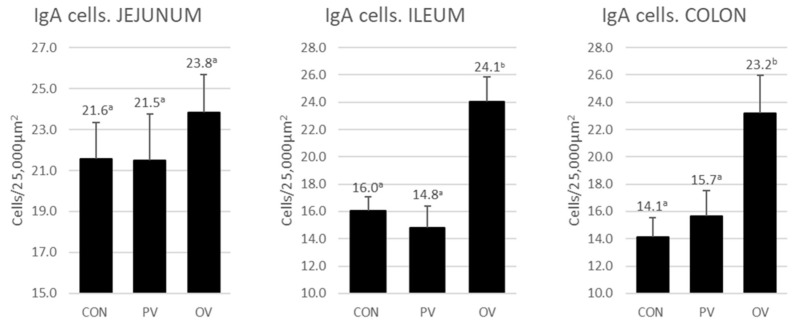
IgA-producing cells in jejunum, ileum and colon for each group. The bars represent the mean ± SEM of cells counted per 25,000 μm^2^. Different superscript in each group indicates significant differences (*p* < 0.005). The data labels are the mean. CON = control group, PV = parenteral vaccine group and OV = oral vaccine group. a b: the same shoulder letters mean no significant difference between groups; different shoulder letters mean significant difference between groups.

**Figure 5 animals-12-02758-f005:**
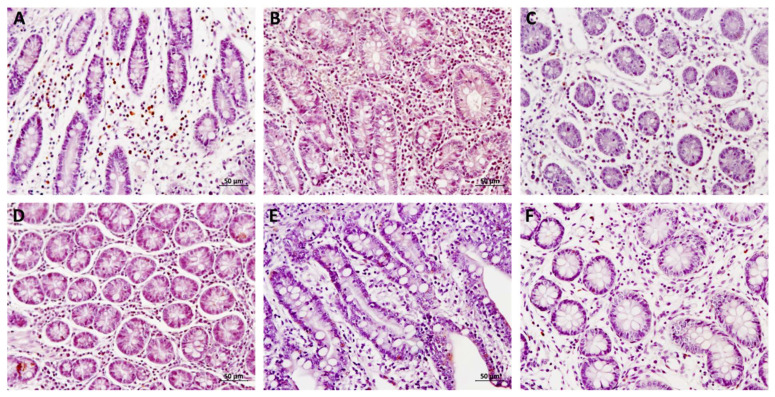
Examples of high density of IgA-producing cells in jejunum (**A**), ileum (**B**) and colon (**C**), and low density for jejunum (**D**), ileum (**E**) and colon (**F**). Hematoxylin–eosin stain, magnification 5×. The bars included are 50 µm.

**Figure 6 animals-12-02758-f006:**
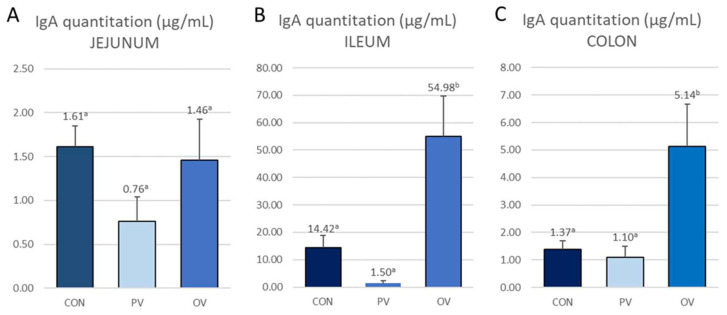
IgA quantitation for (**A**) jejunum, (**B**) ileum and (**C**) colon. The bars show mean ± SEM. The data labels are the mean. CON = control group, PV = parenteral vaccine group and OV = oral vaccine group. a b c : the same shoulder letters mean no significant difference between groups; different shoulder letters mean significant difference between groups.

**Figure 7 animals-12-02758-f007:**
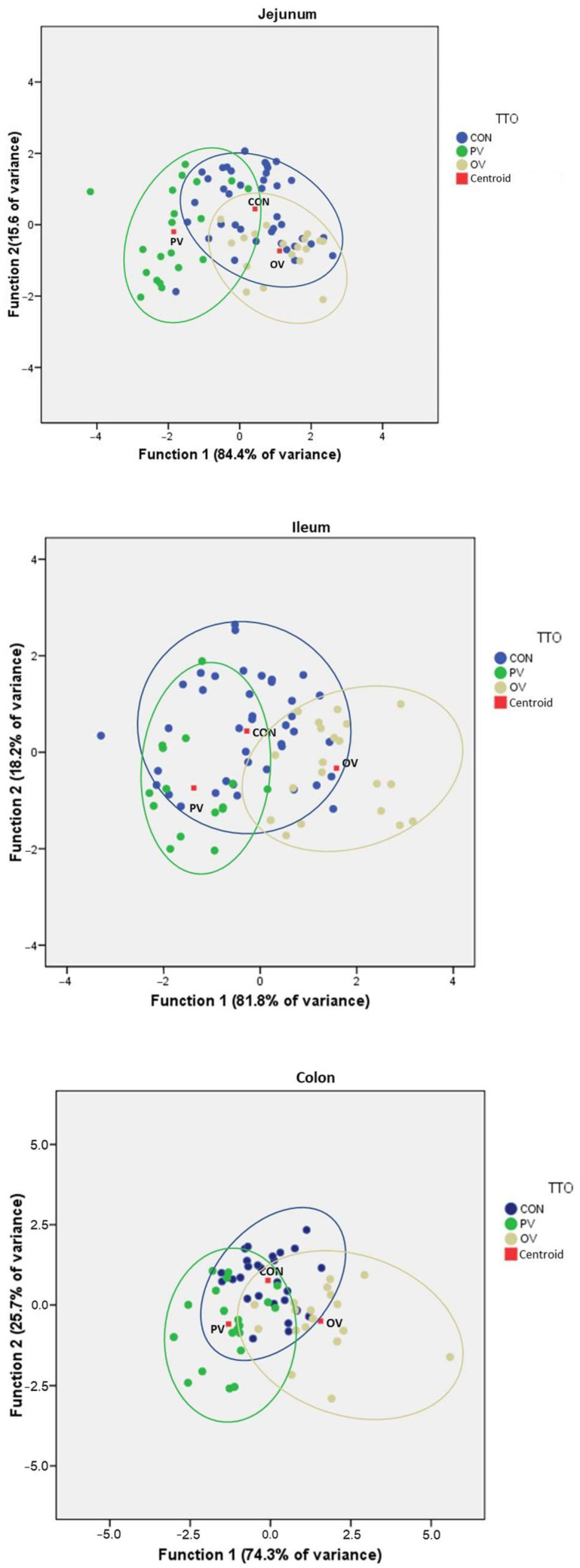
Discriminant functions plot for jejunum, ileum and colon, where the quantifications for biomarkers (CAL, OCL, ZON, IFN-γ and TGF-β) and IgA-producing cells were used. The quantity of variance explained by each function is indicated on the axis of each plot. CON = control group, PV = parenteral vaccine group and OV = oral vaccine group.

**Figure 8 animals-12-02758-f008:**
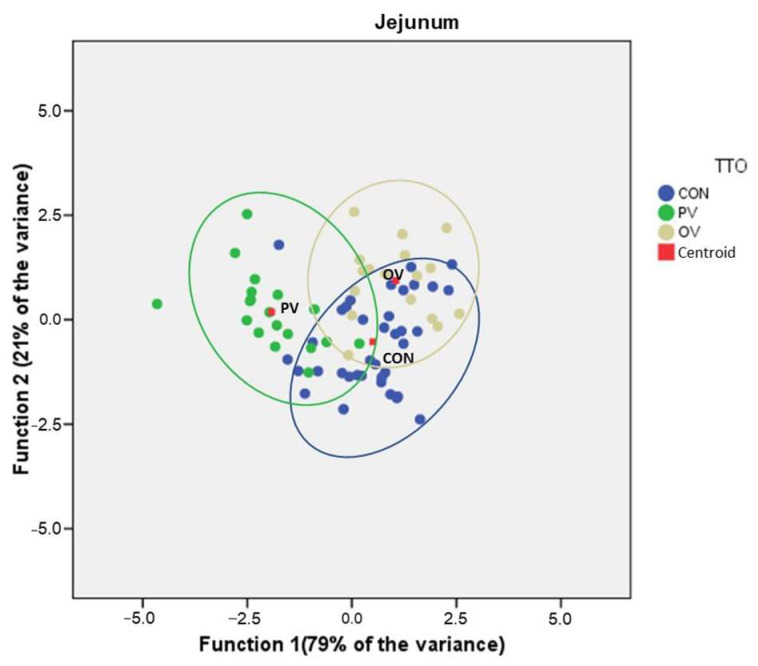
Discriminant functions plot for Jejunum, Ileum and Colon, using the quantifications for biomarkers (CAL, OCL, ZON, IFN-γ and TGF-β), histomorphology parameters (Villi length, Crypt depth and V/C ratio) and IgA-producing cells. The quantity of variance explained by each function is indicated on the axis of each plot. CON = control group, PV = parenteral vaccine group and OV = oral vaccine group.

**Table 1 animals-12-02758-t001:** Primers for TJ proteins cytokines, inflammatory indicator and housekeeper used in this work.

Gene	Forward Primer	Reverse Primer	Accession Number or Reference
CALPROTECTIN (S100 calcium binding protein A8)	5′-AATTACCACGCCATCTACGC-3′	5′-TGATGTCCAGCTCTTTGAACC-3′	NM_001160271.3
*Occludin*	5′-TTGCTGTGAAAACTCGAAGC-3′	5′-CCACTCTCTCCGCATAGTCC-3′	NM_001163647.2
*Zonulin 1*	5′-CACAGATGCCACAGATGACAG-3′	5′-AGTGATAGCGAACCATGTGC-3′	XM_047793702.1
*IFN*-γ	5-TGGTAGCTCTGGGAAACTGAATG-3′	5′-GGCTTTGCGCTGGATCTG-3′	[16]
*TGF*-β	5′-CACGTGGAGCTATACCAGAA-3′	5′-TCCGGTGACATCAAAGGACA-3′	[17]
β-*actin*	5´-CTACGTCGCCCTGGACTTC-3´	5´-GATGCCGCAGGATTCCAT-3´	[18]

**Table 2 animals-12-02758-t002:** Villus height and crypt depth observed for the experimental groups. Different superscript in each row indicates significant differences (*p* < 0.05).

		CON	PV	OV	SEM	*p*-Value
Jejunum	Villus height (µm)	442 ± 15.1 ^a^	510 ± 22.8 ^a^	421 ± 24.0 ^a^	11.21	NS
	Crypt depth (µm)	370.2 ± 16.8 ^a^	380.2 ± 21.1 ^a^	596.6 ± 12.5 ^b^	9.3	0.011
Ileum	Villus height (µm)	363.7 ± 10.4 ^a^	387.5 ± 26.5 ^a^	407.1 ± 15.2 ^b^	8.3	0.003
	Crypt depth (µm)	300.2 ± 11.1 ^a^	324.0 ± 13.3 ^a^	249.9 ± 9.4 ^b^	7.7	0.001

Different superscript in the same row indicates significant differences. CON = control group, PV = parenteral vaccine group and OV = oral vaccine group.

**Table 3 animals-12-02758-t003:** Percentage of samples assigned to each group, and concordance with the predicted belonging group.

Predicted Belonging Group				
Tissue	Percentage	CON	PV	OV
Jejunum	CON	56.4	10.3	33.3
	PV	10.0	90.0	0.0
	OV	27.8	0	72.2
Ileum	CON	60.0	25.0	15.0
	PV	13.3	86.7	0.0
	OV	10.0	5.0	85.0
Colon	CON	74.1	7.4	18.5
	PV	25.0	75.0	0.0
	OV	5.6	5.6	88.9

CON = control group, PV = parenteral vaccine group and OV = oral vaccine group.

**Table 4 animals-12-02758-t004:** Percentage of samples assigned to each group, and concordance with the predicted belonging group.

Predicted Belonging Group				
Tissue	Percentage	CON	PV	OV
Jejunum	CON	70.3	10.8	18.9
	PV	10.5	89.5	0.0
	OV	11.8	0.0	88.2
Ileum	CON	62.2	21.6	16.2
	PV	20.0	80.0	0.0
	OV	10.0	0.0	90.0

## Data Availability

Data are not available since the trial was supported by ELANCO, owner of the data.

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
