# Peer review of "Oral and Parenteral Vaccination against *Escherichia coli* in Piglets Results in Different Responses"

_animals, 2022, doi:10.3390/ani12202758_

Round 1
Reviewer 1 Report
This manuscript is devoted to the evaluation of the “Oral and parenteral vaccination against Escherichia coli in piglets results in different responses”. The authors were trying to compare different routes of administration and to find out which routes of administration could influence “intestinal integrity, cytokines, and intestinal morphology in weaned piglets”. The manuscript is easy to understand, but it needs to be revised according to my comments:
1. Line 119, “LW×LD”, please note this abbreviation. I don’t know LD. Is LW Large White?
2. In the materials and methods, the authors explained that 70 piglets had no clinical problems related to E. coli. However, it was later mentioned that the presence of ETEC F4 was previously detected in rectal swab samples. I think this statement is inconsistent and unreliable. In addition, even if the piglets are healthy in the experiment, the previously existing ETEC F4 can be transmitted through feces or have resistance in pigs, which will affect the effect of vaccination and comparison.
3. (Materials and methods section, line 119) The experimental design used in this study appeared to have some flaws. How many replicates and how many pigs per replicate were in every vaccination treatment? The undefined body weight in experimental piglets was used. Did you adjust the vaccine dosage based on body weight? Whether the measurement parameters were affected by the different initial weaned body weight?
4. Line 139-140. “The piglets were vaccinated at 10th and 20th days of life for PV group, and at weaning for oral vaccine”. The author mentioned that “the piglets were weaned at 22±1 days of life”. Why is the time of the second parenteral injection not at weaning period, but at the 20th day of life? Will this affect the results?
5. Line 143. The authors mentioned that “Thirty piglets were left as controls”. In the following text, the treatment represented by the CON group is not specifically stated.
6. Line 45, (post-weaning diarrhea (PWD)), Line 175, “OCLU”. Wrong abbreviation.
7. Line 55. Non-standard abbreviations should be defined when first used in the text. The European Union (EU).
8. There are also several writing irregularities that need to be revised. For example, line 209, I think “H2O2” should have subscripts. Line 223, Line 231, the "2" in the unit of square micrometer should be a superscript.
9. Figure 1. The figure legends are not self-explanatory. The legends should explain what each number represents. You didn't present anything about your "A, B, C" category in the figure.
10. Table 2. All short explanatory notes in the table should be placed below the table.
11. Figures 2 and 5. The corresponding ruler should be added with every picture. The sample size for histomorphology should be indicated in the legend.
12. A uniform style (p-value) for p-value should be used throughout the whole manuscript.
13. Line 311. The title should be revised to “3.4”. Line 337. The title should be revised to “3.5”.
14. Table 3. Formatted the table according to the requirements of Animals.
15. Please update the reference style according to the guidelines.
16. In the discussion, the reference is missing on lines 449-450, 504-507.
Reviewer 2 Report
The authors found vaccines with different routes of administration results in different responses in piglets. The authors suggest a more rapid and direct effect of oral vaccination and a state of suppression in the absence of a second oral stimulus by the pathogen. The topic is interesting, but appropriate modifications are needed.
Q1. In section 2.1, the initial body weight of piglets should be added in the manuscript.
Q2. In this experiment, why not use western blot assay detect the protein abundance of the tight junction proteins, such as Occludin?
Q3. Could authors describe how the E. coli were cultured and counted.
Q4. In Figure 4, what about the content of IgA.
Q5. Whether different routes of administration affect the intestinal microbiota.
Q6. The authors should measure the serum LPS content.
Q7. The growth performance should be added.
Author Response
Author's NotesThank you very much for your suggestions, which we believe will help to improve the quality of the article. Responding to each specific question:
Q1. In section 2.1, the initial body weight of piglets should be added in the manuscript. Included
Q2. In this experiment, why not use western blot assay detect the protein abundance of the tight junction proteins, such as Occludin?
The determination of TJ proteins by western blot was not included in the experimental protocol.
Q3. Could authors describe how the E. coli were cultured and counted.
The E coli was detected by q-PCR, and were not cultured. It has been included in the text.
Q4. In Figure 4, what about the content of IgA.
The content of IgA have been determined by ELISA and included in a new section
Q5. Whether different routes of administration affect the intestinal microbiota. Was not tested
Q6. The authors should measure the serum LPS content.
Lipopolysaccharide (LPS) endotoxin concentrations have been measured with ELISA kits and reported as potential biomarkers. However, LPS molecules can decrease significantly within one hour after taking the sample, and the interpretation of the results is quite complex, as they are heterogeneous and highly variable depending on the bacteria from which they originated. In addition, LPS molecules detected in plasma could be a tiny proportion of LPS remaining after clearance of bacterial infections and would no longer represent the initial amounts of LPS capable of stimulating host cells. Therefore, LPS concentrations have limited clinical utility (Peek et al., 2004; Senior et al., 2011; Gnauck et al., 2016)
Q7. The growth performance should be added. included
Round 2
Reviewer 1 Report
After being revised by the authors, all my questions have been done.
Author Response
Thank you very much for your review. Your comments and suggestion have been of great value to improve the paper.
Reviewer 2 Report
-
I don't see the authors making any changes to my suggestions.
Author Response
As we have received a third review we consider this revision have been solved.
Thanks
Round 3
Reviewer 2 Report
The protein abundance of the tight junction proteins, such as Occludin is important to show clear and different effects derived from the use of each type of vaccine, route of administration and regimen.
Author Response
Thank you very much for your review again. Specific comments:
Q1: The protein abundance of the tight junction proteins, such as Occludin is important to show clear and different effects derived from the use of each type of vaccine, route of administration and regimen.
Thank you very much for your comment. We agree that quantification of TJ proteins would have enriched the results of this experiment. Unfortunately, it was not included in the initial experimental design, and we do not have a suitable sample for protein quantification and are now unable to perform such analyses. However, we believe it is important that we publish these results as, even in the absence of quantification of these proteins, it provides valuable information on the immune response derived from each type of vaccination, especially as there is no information in the literature so far.
We thank you for your comment, which we consider very valuable, and which would enrich the results.
No doubt, in future experiments related to this topic we will include protein quantification as you suggest.
We hope you understand that it is not that we do not want to follow your recommendations but that we do not have the samples that would allow us to do so.
Thank you very much in advance.
